# Deep Learning in Barrett’s Esophagus Diagnosis: Current Status and Future Directions

**DOI:** 10.3390/bioengineering10111239

**Published:** 2023-10-24

**Authors:** Ruichen Cui, Lei Wang, Lin Lin, Jie Li, Runda Lu, Shixiang Liu, Bowei Liu, Yimin Gu, Hanlu Zhang, Qixin Shang, Longqi Chen, Dong Tian

**Affiliations:** 1Department of Thoracic Surgery, West China Hospital, Sichuan University, 37 Guoxue Alley, Chengdu 610041, China; ruichen_cui@163.com (R.C.); 22wanglei@wchscu.cn (L.W.); linlin@wchscu.cn (L.L.); lijie7109@wchscu.cn (J.L.); runda00@163.com (R.L.); liushixiang@stu.scu.edu.cn (S.L.); liubowei980328@163.com (B.L.); yimin_gu@aliyun.com (Y.G.); drzhanghanlu@wchscu.cn (H.Z.); jiaifang2003@163.com (Q.S.); 2West China School of Nursing, Sichuan University, 37 Guoxue Alley, Chengdu 610041, China

**Keywords:** Barrett’s esophagus, deep learning, diagnosis, endoscope, pathology

## Abstract

Barrett’s esophagus (BE) represents a pre-malignant condition characterized by abnormal cellular proliferation in the distal esophagus. A timely and accurate diagnosis of BE is imperative to prevent its progression to esophageal adenocarcinoma, a malignancy associated with a significantly reduced survival rate. In this digital age, deep learning (DL) has emerged as a powerful tool for medical image analysis and diagnostic applications, showcasing vast potential across various medical disciplines. In this comprehensive review, we meticulously assess 33 primary studies employing varied DL techniques, predominantly featuring convolutional neural networks (CNNs), for the diagnosis and understanding of BE. Our primary focus revolves around evaluating the current applications of DL in BE diagnosis, encompassing tasks such as image segmentation and classification, as well as their potential impact and implications in real-world clinical settings. While the applications of DL in BE diagnosis exhibit promising results, they are not without challenges, such as dataset issues and the “black box” nature of models. We discuss these challenges in the concluding section. Essentially, while DL holds tremendous potential to revolutionize BE diagnosis, addressing these challenges is paramount to harnessing its full capacity and ensuring its widespread application in clinical practice.

## 1. Introduction

Barrett’s esophagus (BE) is a pathological condition where the squamous epithelial cells in the lower part of the esophagus are replaced by a type of columnar epithelium found in the intestine [1,2]. It is often considered a pre-cancerous change for esophageal adenocarcinoma (EAC), with an annual progression rate from BE to EAC estimated to be 0.12–0.13% [3,4]. Early detection and intervention for BE or EAC can result in a five-year survival rate of up to 80%. At the same time, in the late stages, it drops to only 13% [5,6]. Currently, the diagnosis of BE primarily depends on the pathological biopsy under endoscopy. However, even for well-equipped and experienced endoscopists or pathologists, early-stage BE lesions pose a considerable diagnostic challenge [7,8]. Using the Seattle Biopsy Protocol to handle abnormalities detected during endoscopic inspections is commonly recommended. Despite this protocol suggesting four biopsies for every 1 cm, a failure in diagnosis may still occur due to insufficient sample volume [9,10].

Deep learning (DL) has fundamentally changed our approach to analyzing and understanding visual information in recent years. As a subfield of machine learning, the essence of deep learning lies in utilizing neural network models to learn and abstract features from large amounts of data [11,12]. In particular, deep learning has achieved significant breakthroughs in image recognition, analysis, and processing, surpassing human expert levels in specific tasks [13]. In medical image diagnosis, deep learning models can learn typical pathological features from many images, thus potentially identifying BE lesions at an early stage, even those that are minuscule and difficult for the human eye to detect [12,14,15]. Through such means, deep learning may significantly enhance the early diagnosis rate of BE, improving patient prognosis and quality of life.

In this review, we adopted a combined approach of subject keyword and free word search strategy to identify pertinent studies focusing on deep learning and Barrett’s esophagus. Our primary data sources encompassed four public databases: PubMed, Embase, Web of Science, and Cochrane Library. Using a predefined set of inclusion and exclusion criteria, we meticulously assessed the literature from the past decade. Inclusion Criteria: 1. Studies primarily addressing the application of deep learning in Barrett’s esophagus diagnosis. 2. Articles presenting original data and specific research findings. 3. Publications within the last ten years. Exclusion Criteria: 1. Duplicates or multiple versions of the same study. 2. Commentaries, expert opinions, case reports, or any non-original research articles. 3. Studies not directly relevant to deep learning or Barrett’s esophagus. Following this rigorous screening process, we distilled our initial findings from 213 articles down to 33 original studies for an in-depth analysis (Figure 1). Through this review, we offer an exhaustive insight into the current applications of deep learning in BE diagnosis and further discuss its promising trajectory in an upcoming clinical setting.

## 2. Application of Deep Learning to Assist Endoscopic Diagnosis

Endoscopy plays a pivotal role in diagnosing BE, enabling gastroenterologists to view the entirety of the esophageal inner wall and sample specific tissues for pathological examination based on the situation. A comprehensive review included 21 original studies that utilized endoscopic images or video information to construct deep neural networks to aid diagnosis (Table 1).

Diagnosing diseases first considers the distinction between the pathological site and the surrounding normal tissues. Liu, G., et al. constructed and verified a convolutional neural network (CNN) with two subnetworks using 1272 white-light endoscopic images from a single-center retrospective study, completing the classification task of normal vs. pre-cancer vs. cancer. The final model performance showed an accuracy of 85.83%, a sensitivity of 94.23%, and a specificity of 94.67%. One subnetwork was designed to extract color and global features. In contrast, the other extracted texture and detailed features to increase the model’s interpretability [25]. Leandro A. Passos and his team used the infinite restricted Boltzmann machines to classify esophageal lesions (BE) from other conditions based on the publicly available endoscopy images from the “MICCAI 2015 EndoVis Challenge” dataset. In this study, the maximum accuracy reached 67% [19]. In theory, video provides more dimensional information than images, potentially enhancing prediction effectiveness. Pulido, J.V., et al. retrospectively collected 1057 probe-based confocal laser endomicroscopy (pCLE) videos from 78 patients and constructed models based on AttnPooling and MultiAttnPooling architecture, completing the classification task of normal vs. non-dysplastic BE (NDBE) vs. dysplastic BE/cancer. The final model performance for AttnPooling showed a sensitivity of 90% and a specificity of 88%, whereas for MultiAttnPooling, the sensitivity and specificity were 92% and 84%, respectively [28]. Van der Putten J. and colleagues discovered that during the process of creating endoscopic videos, there might be some invalid frames due to various reasons, affecting lesion detection. Therefore, a method was proposed that initializes frame-based classification and then employs the Hidden Markov Model to incorporate temporal information, enhancing the sensitivity of the classification by 10% [20].

Apart from distinguishing normal tissues from BE, some researchers have also paid attention to differentiating esophagitis from BE. Kumar, A.C., et al. attempted combinations of five CNN architectures and six classifiers using 1663 endoscopic images from public databases. The best-performing model was a fine-tuned ResNet50 with transfer learning, achieving an AUC of 0.962 [35]. Additionally, Villagrana-Bañuelos, K.E., et al. subdivided esophagitis and accomplished a four-classification task: normal vs. BE vs. esophagitis-a vs. esophagitis-b-d. They used 1561 endoscopic images from public databases to construct a model based on the VGG architecture, with the final model’s AUC being normal: 0.95, BE: 0.96, esophagitis-a: 0.86, and esophagitis-b-d: 0.83 [36].

Since BE is a known precursor to EAC, the transition from BE to EAC is a clinical concern. Ebigbo, A., et al. utilized 248 endoscopic images from a public database to classify BE and EAC. Using the ResNet model, they achieved a sensitivity of 97% and a specificity of 88% on the Augsburg dataset, and a sensitivity of 92% and a specificity of 100% on the MICCAI dataset [18]. Ghatwary, N., et al. aimed to build an object detection model based on endoscopic images. Through trials of multiple models, they ultimately chose the VGG-based single-shot multibox detector as the prediction model. The model was trained on endoscopic images from a public database of 100 cases, completing the BE vs. EAC classification task and achieving a sensitivity of 96% and a specificity of 92% [22]. Hou, W., et al. proposed an end-to-end network with an attentive hierarchical aggregation module and a self-distillation mechanism, achieving an AUC of 0.9629. This model enhances classification performance without sacrificing temporal performance, thereby achieving real-time inference [31]. Additionally, de Souza, L.A., Jr., et al. constructed a model for the same task using a convolutional neural network combined with a generative adversarial network (GAN). Their choice of using GANs was driven by the challenges related to limited datasets in medical imaging and the potential of GANs in data augmentation, especially for generating synthetic yet realistic medical images. This approach achieved an accuracy of 90% for the patch-based method and 85% for the image-based approach [24]. The pursuit of creating explainable artificial intelligence continues to be an ongoing endeavor that tracks the operational process of deep learning technology and offers insights into the correctness or error behind its models. De Souza, L.A., Jr., et al. used five different explanation techniques (saliency, guided backpropagation, integrated gradients, input × gradients, and DeepLIFT) to analyze four commonly used deep neural networks (AlexNet, SqueezeNet, ResNet50, and VGG16) built for BE vs. EAC classification tasks through endoscopic images, demonstrating the correlation between computational learning and expert insights [33].

From the perspective of BE subclasses, some researchers have modeled and predicted whether there are developmental abnormalities. Several researchers have tried to model predictions for the presence or absence of dysplasia within the subcategories of BE. Van der Putten, J., et al. collected prospective data sets from three centers. They constructed a model using the ResNet architecture with 40 endoscopic images, thereby completing the classification task of neoplastic BE vs. NDBE. The model achieved a final accuracy of 98%. Its innovation lies in the fact that the deep learning model can assist clinicians in determining the optimal location for pathological biopsies [21]. The team published another phased model for the same problem in the same year. In the first phase, the extraction of features was completed through an editor structure, and in the second phase, the classification task of NDBE vs. dysplastic BE was achieved through a ResNet-based structure. They explored the effect of pre-training on model performance and which dataset would provide the best performance when used for pre-training. The results indicated that pre-training based on the GastroNet dataset yielded the best performance, with an AUC reaching 0.91 (the other two were 0.82 and 0.90, respectively) [23]. The following year, van der Putten, J., et al. incorporated five datasets for modeling, aiming to address this problem better. The T1 dataset contained 494,355 gastrointestinal organ images for pre-training; the T2 dataset consisted of 1247 endoscopic images for model training; the T3 dataset, containing 297 endoscopic images, was used for model parameter adjustment; and finally, the T4 and T5 datasets, together composed of 160 endoscopic images, served as an external test set to evaluate model performance. The multi-stage model they constructed completed the segmentation task using a U-Net architecture and the classification task using a ResNet architecture, ultimately achieving an accuracy, sensitivity, and specificity of 90% [27]. The ultimate purpose of model construction is to serve clinical practice. De Groof AJ and others built a classification model for NDBE vs. neoplastic BE based on the ResNet/U-Net architecture model. They validated it using a prospective, multicenter dataset, achieving an accuracy of 90%, sensitivity of 91%, and specificity of 89%. Worth mentioning is that this team attempted to apply the model to clinical practice with 20 patients after its construction to demonstrate the consistency of computer-assisted detection (CAD) predictions and diagnoses [26]. Jisu, H., et al. conducted a study based on 262 endoscopic images targeting subtypes with BE. Using a conventional CNN model, they accomplished a tri-classification task of intestinal metaplasia, gastric metaplasia, and neoplasia, achieving an accuracy of 80.77% [16].

Compared to standard endoscopy, narrow-band imaging (NBI) endoscopy enhances the visualization of the mucosa and vasculature [37,38,39,40]. Struyvenberg, M.R., et al. initially conducted pre-training using the GastroNet database. Subsequently, they trained using another dataset composed of 1247 white-light endoscopic images. Following this, they employed the third dataset containing NBI endoscopic images, undertaking additional training and validation within an internal center. Lastly, they used the fourth dataset, composed of NBI videos, for external validation. When performing the classification task of neoplastic BE vs. NDBE, the prediction results for image information were a sensitivity of 88% and a specificity of 78%. For video information, the prediction results were a sensitivity of 85% and a specificity of 83% [29]. Additionally, Kusters, C.H.J., et al., based on the EfficientNet-b4 architecture, performed predictions for NBI data using datasets from seven centers. They conducted training on image data and validated and tested it on video data. For completing the neoplastic BE vs. NDBE classification task, the highest achieved AUC value was 0.985 [34].

Determining the extent of BE lesions is crucial for diagnosis, treatment monitoring, and prognosis prediction [41,42]. Pan, W., et al., based on the architecture of fully convolutional networks, constructed a model to accomplish the segmentation task of delineating the boundary between BE and normal esophageal epithelium, utilizing 443 endoscopic images from a single-center prospective study. The performance of their model was evaluated by the intersection over union: 0.56 (at the gastroesophageal junction) and 0.82 (at the squamous-columnar junction) [30]. De Groof J. and others conducted esophageal lesion boundary detection using 40 endoscopic images. They used a pre-trained Inception v3 to build the model, and the final segmentation score was 47.5%. Although the performance was not as good as that of human experts, this study focused on the detection of esophageal lesion boundaries [17]. Furthermore, Ali, S., et al. created an automated model for the quantification and segmentation of BE using endoscopic images and videos, with an accuracy of 98.4% [32].

These studies employ deep learning technologies based on endoscopic images or video information to diagnose BE. They primarily cover three key areas: disease classification, severity grading of disease, and lesion segmentation. They provide auxiliary diagnostic means with significant potential for clinical application.

## 3. Applications of Deep Learning to Assist Pathological Diagnosis

With the rapid advancements in digitization and artificial intelligence, digital pathology has emerged as an integral component of modern medicine, providing clinicians with a more precise and quicker method of diagnosis [43,44,45,46]. At its core, digital pathology utilizes complex algorithms and big data to analyze images of pathological tissues, aiming to identify and assess lesions. In the diagnosis of BE, digital pathology is increasingly prevalent, with a series of research studies deeply exploring this application [47,48]. The primary objective of these studies is to leverage deep learning technologies for more accurate identification and categorization of BE-related lesions, thereby enhancing the precision and timeliness of BE diagnosis [49,50]. In the following sections, we will discuss the specific content of seven research studies (Table 2).

Pathology represents the gold standard for diagnosis, potentially offering more comprehensive and microscopic histological information in theory than endoscopy. As the precursor of EAC, BE presents a unique opportunity to understand the factors driving the transition from pre-cancerous conditions to cancer [3,58]. Cell detection is usually the first crucial step in the automated image analysis of pathological slides [59,60]. Law, J., et al. developed a model based on SE2-U-Net, using pathological images from multiple public datasets to classify cells and non-cells in pathological sections [53].

Subsequently, the task of pathological diagnosis and classification comes into play. Tomita, N., et al. built a deep learning network model based on an attention-driven CNN. They used 123 retrospective case images from a single center for training, validation, and prediction, accomplishing a four-category task: normal vs. NDBE vs. dysplastic BE vs. EAC. The final model performance indicators were: AUC for normal: 0.751; NDBE: 0.897; dysplastic BE: 0.817; and EAC: 0.795 [51]. This model can accurately identify cancerous and pre-cancerous esophageal tissues on microscopic images without training annotations for areas of interest, significantly reducing the workload in pathological image analysis. Sali, R., et al. used 387 pathological images to construct a model based on the ResNet architecture to accomplish a three-category task: normal vs. NDBE vs. dysplastic BE. During the experiments, they compared the impact of full supervision, weak supervision, and unsupervised learning on model performance. The results indicated that the appropriate setting of unsupervised feature representation methods could extract more relevant image features from whole-slide images [52]. Guleria, S., et al. conducted a modeling study where they compared pCLE with pathological images. Their findings demonstrated that a comprehensive three-category classification task, distinguishing between normal, NDBE, and dysplasia/cancer, can be successfully achieved using multiple retrospective datasets [57].

Moreover, the risk of BE transforming into EAC increases with the progression of atypical hyperplasia. However, the histological diagnosis of atypical hyperplasia, particularly low-grade atypical hyperplasia, presents challenges, leading to a lack of inter-observer consensus among pathologists [61]. Therefore, Codipilly, D.C., et al. used 587 patient pathological slices to build a model based on the ResNet architecture, accomplishing a three-category task: NDBE vs. low-grade dysplasia vs. high-grade dysplasia [54]. In the same task, Faghani, S., et al. constructed a two-stage model, with the first stage identifying and segmenting using YOLO [56]. The output of the first stage was then used as input for the second stage, which, in conjunction with the ResNet architecture, completed the three-category task [56].

Mass spectrometry imaging (MSI) can obtain spatially resolved molecular spectra from tissue slices without labeling. Beuque, M., et al. combined MSI with standard pathological slice data to construct a model with three modules that can complete tasks: Task 1: epithelial vs. stroma; Task 2: dysplastic grade; and Task 3: progression of dysplasia. The inclusion of more information may contribute to a more precise classification [55].

The application of deep learning to aid pathological diagnosis has been extensively studied. Research focuses on utilizing complex algorithms and big data to analyze pathological tissue images for more accurate and timely identification and classification of lesions. Various studies have made progress, including cell vs. non-cell classification, multi-category tasks for BE, and more precise diagnoses integrating mass spectrometry imaging. These research findings provide robust support for improving the accuracy and efficiency of pathological diagnosis.

## 4. Applications of Deep Learning to Assist Other Diagnostic Methods

In addition to traditional endoscopy and conventional pathological sampling, some researchers have developed deep neural network models for other diagnostic methods (Table 3). Volumetric laser endomicroscopy (VLE) is a novel imaging technique. During the examination, a balloon is inflated in the esophagus, and second-generation optical coherence tomography (OCT) is used to capture a full circumferential scan of the esophageal wall (approximately 6 cm) in about 90 s, reaching a depth of up to 3 mm [62]. This method can acquire information from deep tissues, but interpreting the gray-shadow images is challenging for human experts. To address this, Fonollà, et al. used VLE image data to distinguish between NDBE and HGD using the deep learning model VGG-16, achieving a maximum AUC value of 0.96 [63]. Van der Putten, J., et al., utilizing prospective single-center data and based on the FusionNet/DenseNet architecture, effectively completed a binary classification task of NDBE vs. HGD, achieving an AUC of 0.93 [64]. Additionally, based on OCT information, Z. Yang and others proposed a bilateral connectivity-based neural network for in vivo human esophageal OCT layer segmentation. This marked the development of the first end-to-end learning method specifically designed for automatic epithelial cell segmentation in vivo in human esophageal OCT images [65].

An exciting study was conducted by Gehrung, M., et al., in which an endoscopic brush-like sponge was used to collect epithelial cells. H&E staining and immunohistochemical staining of trefoil factor 3 were performed to obtain the image information input for the model. The model, built based on the VGG-16 architecture, was used to distinguish between BE and others, with an AUC reaching up to 0.88 [66]. Other researchers discovered that when light passes through tissue, it is absorbed by endogenous chromophores such as hemoglobin and scattered by endogenous structures such as organelles and nuclei. Biochemical and structural changes associated with diseases in the epithelial layer alter the distribution and abundance of absorbers and scatterers, leading to subtle changes in the spectral characteristics of the light exiting the tissue. Spectral imaging techniques can capture this rich endogenous contrast to reveal potential pathological changes [68,69]. Waterhouse, D.J., et al. modeled the spectral signals and completed the NDBE vs. EAC classification task with prospective data. The first clinical human trial demonstrated the potential of spectral endoscopy to reveal disease-related vascular changes and provide a highly contrasting depiction of esophageal tumor formation [67].

Compared to human experts, artificial intelligence has a more remarkable ability to capture patterns in large datasets, especially the potential information between data that is hard to discern with the naked eye. Therefore, one possible future research direction is to focus on collecting more information through various means and using it to assist in diagnosis.

## 5. Public Databases and Model Evaluation Metrics

In the literature we reviewed, the commonly used public datasets mainly include the following: Firstly, the HyperKvasir dataset was collected at Bærum Hospital in Norway and is derived from gastroscopy and colonoscopy examinations. Part of the data has been annotated by experienced gastroendoscopy physicians. This dataset covers 110,079 images and 374 videos, showcasing anatomical landmarks, various pathological conditions, and normal findings. Secondly, GastroNet is composed of 494,364 endoscopic images from 15,286 patients, covering organs including the colon, stomach, duodenum, and esophagus. Thirdly, the ImageNet database encompasses over 14 million manually annotated high-resolution images, covering more than 22,000 categories of objects and entities. Fourthly, the MICCAI 2015 dataset includes 100 lower esophageal endoscopic images captured from 39 individuals. Out of them, 22 were diagnosed with Barrett’s esophagus, and 17 showed early signs of esophageal adenocarcinoma. Fifthly, the Augsburg dataset consists of 76 endoscopic images from patients with BE (42 samples) and early adenocarcinoma (34 samples). These databases have significant application value in the fields of medical imaging and computer vision research. Specifically, HyperKvasir, GastroNet, and ImageNet are frequently used as sources for pre-training models, while MICCAI 2015 and Augsburg are often utilized in formal model training and external validation.

In the model performance evaluations mentioned above, for classification tasks, accuracy, sensitivity, specificity, F1 score, and area under the receiver operating characteristic curve are commonly employed as evaluation metrics. For segmentation tasks, aside from accuracy, some researchers also use metrics such as intersection over union and dice coefficient to assess segmentation performance.

## 6. Discussion

In this review, we conducted a literature search across multiple databases, culminating in the inclusion of 33 primary studies. The objective was to summarize the current role of deep learning in aiding the diagnosis of BE. We identified that the types of data for modeling can be categorized as discussed herein. First, endoscopic data, including images and videos, are used for prediction modeling. Second, pathological images (e.g., H&E staining or IHC) are modeled for prediction. Third, other auxiliary diagnostic information (such as OCT) is utilized for predictive modeling. Regarding the types of diagnostic tasks, there are two primary categories: classification tasks, which encompass binary and ternary classification tasks distinguishing BE from normal tissues or tumorous tissues, and segmentation tasks, which focus on the segmentation of epithelial tissues or individual cells (Figure 2A). In some of the aforementioned studies, researchers delineated segmentation and classification tasks into two distinct phases. Initially, problematic esophagi are segmented, followed by classification tasks targeting the segmented esophagi. This phased model approach deconstructs a complex task into multiple stages, each employing a specific model to address sub-problems, sequentially accomplishing the entire task. In every stage, a dedicated model processes the data, producing intermediary outcomes, which are then forwarded to the subsequent phase’s model for further processing until the entire task is finalized [70,71,72]. 

Given the outcomes of these 33 studies, all exhibited commendable model performance. Consequently, we believe that deep learning holds promising potential for augmenting the diagnosis of BE.

From an architectural perspective, researchers chose different foundational model architectures for various tasks or data types and made adaptive improvements (Figure 2B). For classification tasks, image data was predominantly processed using CNN architectures such as VGG, AlexNet, and ResNet. AlexNet, designed in 2012, was pivotal in the ImageNet image recognition challenge, establishing the efficacy of deep convolutional neural networks in large-scale image classification tasks [73]. In this BE diagnostic review, studies by de Souza LA Jr. and Kumar A. C. have attempted to employ this architecture [33,35]. VGG, proposed by the Oxford Vision Group in 2014, secured the second position in the ImageNet challenge. Despite not clinching the title, VGG showcased the potential of deep convolutional networks, particularly through increased depth [74]. This BE review encompasses four studies involving this architecture [22,33,36,63,66]. ResNet, designed by Microsoft Research in 2015, became widely adopted due to its residual network design, which addressed the vanishing gradient problem in deep network training [75]. This BE review indicates 12 studies utilizing ResNet, revealing a preference for its stable performance in classification tasks [18,20,21,23,26,27,29,31,33,35,52,54,55,56]. DenseNet, introduced by Cornell University researchers in 2017, embraced dense connections to enhance feature reuse and gradient flow [76]. In this BE review, research by van der Putten J. and Kumar A. C. tried this architecture [35,64]. For segmentation tasks, most adopted the U-Net architecture with modifications. U-Net, proposed in 2015 by a German image processing institute, was specifically designed for biomedical image segmentation [77]. This review found four studies using U-Net [26,27,29,53]. We argue that modeling should not be confined to one architecture but should explore diverse methods, opting for the most efficacious model, as exemplified by Kumar A. C. and colleagues [35].

The introduction of deep learning in medical diagnosis has brought revolutionary changes to the field. However, we must clarify its actual application intent: as a supplementary tool to complement and enhance doctors’ expertise, not to completely replace them. Human intuition, experience, and years of training cannot simply be replaced by machines. On the contrary, deep learning should be viewed as a tool aimed at providing more accurate and faster data analysis to assist doctors in making better decisions. In our review, we noted that some researchers have made comparisons between deep learning models and human experts. The results show that, especially on certain metrics, the performance of deep learning models is comparable to expert diagnosis (Table 4). This has mainly been verified in scenarios where tasks are clear and the data set quality is high. This further proves the potential value of deep learning in the medical field. For instance, for some complex image analysis tasks, models can quickly identify potential abnormal areas, helping doctors narrow the scope of examination.

Although deep learning has achieved remarkable strides in diagnosing Barrett’s esophagus, we must recognize the inherent challenges and shortcomings. In the subsequent sections, we delve deeper into these shortcomings and explore potential countermeasures for more reliable diagnostic methods. Primarily, the model’s performance could be hampered by data limitations and inherent model characteristics. Most researchers utilized retrospective single-center data, possibly leading to overfitting and thus compromising generalization. To address this, we recommend prospective multi-center joint data for training and validation to ensure data quality and diversity. However, such data collaboration also brings a series of challenges. First, there are issues of data privacy and security. Medical data often contains sensitive information about patients. When sharing data across centers, it is essential to ensure that this data is not misused or leaked. To address this, researchers can consider using techniques such as data de-identification and anonymization to ensure privacy and security during the data-sharing process. Secondly, there is the challenge of data heterogeneity. Different centers may adopt various standards for data collection, storage, and processing, leading to data heterogeneity that might impact model performance. To resolve this issue, normalization and standardization must be performed before integrating the data, ensuring data consistency. Third, there are legal and ethical concerns related to data sharing. Beyond technical issues, data sharing encompasses multiple legal and ethical considerations. This necessitates that researchers obtain appropriate ethical review and patient consent before sharing data, ensuring compliance with relevant legal stipulations.

Secondly, in the application of multidimensional data modeling, there is limited utilization in this direction. In fact, multidimensional information modeling has demonstrated potential for enhancing the predictive performance of clinical models across various fields. This modeling approach integrates multiple data features and information, aiming to construct a more comprehensive and accurate model [78,79,80]. It can take into account factors such as a patient’s medical history or multiple related examinations for classification purposes. Successful applications have been observed in areas such as lung cancer and breast cancer [81,82,83]. We believe this will be a promising research direction.

Thirdly, it is undeniable that the quality of the endoscopic system influences the final image quality, which in turn affects the accuracy of deep learning models. High-resolution sensors are capable of capturing more details, thereby providing richer information for deep learning models. At the same time, sensors with a wide dynamic range ensure that clear images can be obtained under various lighting conditions. A high-quality illumination system ensures that both doctors and algorithms can clearly see every detail of the tissue. Uniform, shadow-free illumination helps emphasize abnormal areas, making it easier for the model to detect lesions. Insufficient or uneven lighting might obscure crucial information or lead to color distortions in the image. In the studies we reviewed, few provided detailed information about the technical specifications of their endoscopic systems. This omission might be a potential limitation because different systems might produce varying image qualities, leading to differences in model performance. Therefore, we recommend that in future research, researchers should explicitly provide detailed information about the technical specifications of the endoscopic system. This not only enhances the transparency of the research but also helps to better understand under which equipment conditions the performance of deep learning models is optimal.

Furthermore, the interpretability of deep learning models remains an immensely challenging field. Deep learning models are often regarded as “black boxes,” which constitutes a significant barrier to the clinical application of deep neural networks [84,85]. Despite their impressive performance in Barrett’s esophagus diagnosis, the internal mechanisms of these deep learning models are frequently exceptionally intricate, making it difficult to explain the basis for the model’s decisions. In the realm of medical diagnosis, interpretability becomes especially critical, as both medical practitioners and patients require an understanding of how the model arrives at its diagnostic conclusions. It is worth mentioning that although de Souza LA Jr. and colleagues attempted five distinct explanation techniques in their research, including saliency, guided backpropagation, integrated gradients, input time gradients, and DeepLIFT, this undertaking was a proactive effort [33]. Nonetheless, conquering this issue still appears to be a challenge.

## 7. Conclusions

Deep learning has played a pivotal role and demonstrated tremendous potential in diagnosing BE (Figure 3). Its applications range widely, from primary image classification to more complex segmentation tasks and advanced lesion detection, where deep learning has shown powerful capabilities. Leveraging advanced network architectures such as U-Net, ResNet, etc., deep learning has achieved remarkable success in big data processing, pattern recognition, and precise localization. Moreover, some studies have gone even further, using deep learning techniques for spectral image analysis or OCT image information to uncover more potential pathological changes, which brings new possibilities for early diagnosis of BE. However, despite the significant advancements made by deep learning in diagnosing BE, challenges remain that need to be addressed and resolved, particularly those related to model interpretability and credibility. Nevertheless, with continuous algorithm optimization and the application of new technologies, such as advanced model explanation techniques, there is reason to believe that deep learning will play an even more significant role in the diagnosis and research of BE. This will enable us to provide more accurate, earlier diagnoses, ultimately leading to better treatment options and quality of life for patients.

## Figures and Tables

**Figure 1 bioengineering-10-01239-f001:**
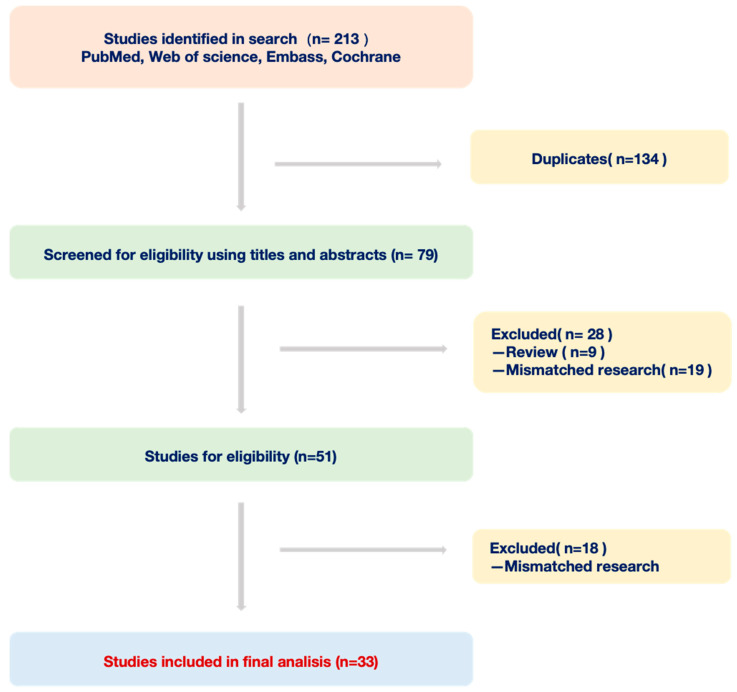
Workflow of documents from four databases.

**Figure 2 bioengineering-10-01239-f002:**
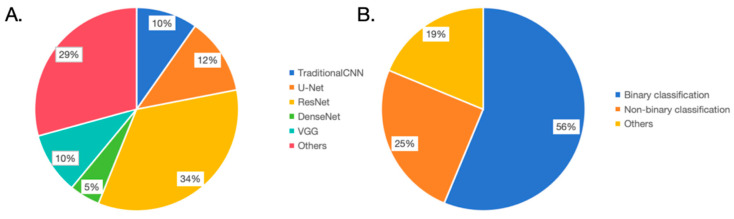
Overview of the model structure and task types. (**A**). In terms of model structure usage frequency, U-Net: 34%, VGG: 29%, Traditional CNN: 12%, ResNet: 10%, DenseNet: 5%, and others: 10%. (**B**). In task types, 56% of the studies focus on binary classification, 25% on non-binary classification, and 19% on other task types.

**Figure 3 bioengineering-10-01239-f003:**
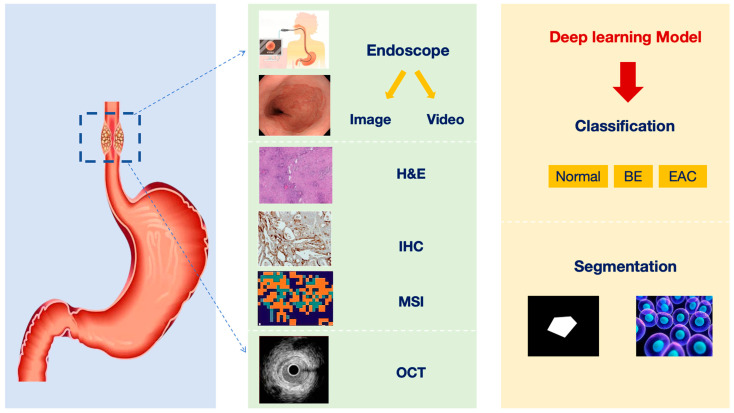
Overview of the current status of deep learning technology-assisted diagnosis of BE (H&E: hematoxylin-eosin; IHC: immunohistochemistry; MSI: mass spectrometry imaging; OCT: optical coherence tomography; BE: Barrett’s esophagus; EAC: esophageal adenocarcinoma).

**Table 1 bioengineering-10-01239-t001:** Twenty-one studies on deep learning-assisted endoscopic diagnosis of BE.

Author	Year	Task	Dataset Size	Data Type	Methodology and Innovation	Model Architecture	Comparison with Experts	Result
Jisu, H. [16]	2017	Intestinal metaplasia vs. Gastric metaplasia vs. Neoplasia	262	Endoscopic images	Augmented Data, Iterations: 15,000, Batch: 20, Optimizer: Adaptive Subgradient, LR: 1 × 10^−5^, LR Decay: 1/10 per step	Traditional CNN	No	Accuracy: 80.77%
de Groof, J. [17]	2018	BE boundary detection	40	Endoscopic images	NA	Inception v3	Yes	Delineation scores: 47.5%
Ebigbo, A. [18]	2019	BE vs. EAC	248	Endoscopic images	Leave-one-patient-out cross-validation, Augmented Patches from Endoscopic Images	ResNet	No	Sensitivities/specificities of 97%/88% (Augsburg data); Sensitivities/specificities of 92%/100% (MICCAI data)
Passos, L.A. [19]	2019	BE vs. Others	100	Endoscopic images	Adopts 6 meta-heuristic technology optimization algorithms	Infinite restricted Boltzmann machines	No	MAX accuracy: 67%
van der Putten, J. [20]	2019	Normal vs. BE	86	Endoscopic videos	Using a pre-trained deep learning model and the Hidden Markov Model to automatically classify endoscopic video frames as ‘informative’ or ‘non-informative’	ResNet	No	Accuracy: 94% Sensitivity: 86% Specificity: 96%
van der Putten, J. [21]	2019	NDBE vs. Neoplastic BE	40	Endoscopic images	Utilize two-stage training with differential learning rates to optimize transfer learning from ImageNet, and during post-processing, average predictions over multiple image transformations and define lesion areas based on a score threshold	ResNet	Yes	Accuracy: 98% Sensitivity: 100% Specificity: 95%
Ghatwary, N. [22]	2019	BE vs. EAC	100	Endoscopic images	Utilize a range of deep learning-based object detection methods during modeling, including R-CNN, Fast R-CNN, Faster R-CNN, and SSD, to detect EAC	Single-shot multibox detector based on VGG	No	Sensitivity: 96% Specificity: 92%
van der Putten, J. [23]	2019	NDBE vs. Dysplastic BE	Pre-training: 494,364 Two centers: 159	Endoscopic images and videos	Use endoscopic imagery for pre-training and validate using four-fold subject-wise cross-validation	Encoder/ResNet	No	AUC: 0.91
de Souza, L.A., Jr. [24]	2020	BE vs. EAC	Public repository	Endoscopic images	Adopt patch- and image-based preprocessing strategies, apply data augmentation, and perform 20-fold cross-validation	CNN based on a generative adversarial network	No	Accuracy: 90% patch-based approach Accuracy: 85% image-based approach
Liu, G. [25]	2020	Normal vs. Precancer vs. Cancer	1272	Endoscopic images	Contrast-enhanced esophageal images were used as input to the CNN and trained using data augmentation and a two-stream CNN algorithm, combining features of the original and pre-processed images	CNN with two subnetworks	No	Accuracy: 85.83% Sensitivity: 94.23% Specificity: 94.67%
de Groof, A.J. [26]	2020	NDBE vs. Neoplastic BE	Pre-training: 494,364 Training: 1544 Test: 160 Application: 20	Endoscopic images	Use a pre-trained large dataset to initialize the deep learning CAD system, then fine-tune with Barrett’s epithelium-specific images, employing a custom hybrid model for simultaneous image classification and segmentation	ResNet/U-Net	No	Accuracy: 90% Sensitivity: 91% Specificity: 89%
van der Putten, J. [27]	2020	NDBE vs. Dysplastic BE	T1 pre-training: 494,355 T2 training: 1247 T3 validation: 297 T4 + T5 test: 160	Endoscopic images	Developed a computer-aided classification and localization algorithm using a semi-supervised learning approach and optimized it through a multi-stage transfer learning strategy	U-Net/ResNet	Yes	Accuracy: 90% Sensitivity: 90% Specificity: 90%
Pulido, J.V. [28]	2020	Normal vs. NDBE vs. Dysplastic BE/Cancer	1057	Endoscopic videos	The video classification model includes frame-level networks, pooling networks, and classifiers, using attention pooling technology to highlight the importance of each frame in video classification	AttnPooling/MultiAttnPooling	No	AttnPooling: sensitivity: 90%, specificity: 88% MultiAttnPooling: sensitivity: 92%, specificity: 84%
Struyvenberg, M.R. [29]	2021	NDBE vs. Neoplastic BE	Pre-training: 494,364 Endoscopic images: 1247 NBI images: 183 NBI videos: 157	Endoscopic images and videos	NBI is trained on still images and improves performance through automatic video analysis, taking the average prediction of all frames within the video	Resnet/U-Net	No	Image: sensitivity 88%, specificity 78% video: sensitivity 85%, specificity 83%
Pan, W. [30]	2021	BE and normal tissue segmentation	443	Endoscopic images	Extract the feature map of the input image through a multi-layer convolutional network and achieve pixel-level semantic segmentation	FCN	No	Intersection over union: 0.56 (GEJ), 0.82 (SCJ)
Hou, W. [31]	2021	BE: Cancer vs. No-cancer	100	Endoscopic images	Proposed a novel end-to-end network equipped with an attention hierarchical aggregation module and self-distillation mechanism	SE-ResNet50	No	AUC: 0.9629
Ali, S. [32]	2021	Automatically quantify Barrett’s epithelium	131	Endoscopic images and videos	Automatically quantify Barrett’s epithelium and measure Barrett’s length and Barrett’s area	NA	No	Accuracy: 98.4%
de Souza, L.A., Jr. [33]	2021	BE vs. EAC	176	Endoscopic images	Four convolutional neural network models were analyzed using five different interpretation techniques to compare their consistency with expert previous annotations of cancer tissue	AlexNet/SqueezeNet/ResNet/VGG	No	Explain the “black box”
Kusters, C.H.J. [34]	2022	NDBE vs. Neoplastic BE	Images: 1748 Neoplastic BE, 1762 NDBE Videos: 90 Neoplastic BE, 194 NDBE	Endoscopic images and videos	Build an endoscope-driven, pre-trained deep learning-based model to characterize NBI images of BE and evaluate the algorithm’s performance on images and videos	EfficientNet-b4	No	AUC: 0.985
Kumar, A.C. [35]	2022	Esophagitis vs. BE	1663	Endoscopic images	Try as many model frameworks and classifier combinations as possible to find the optimal model	5 CNN structures and 6 classifiers	No	MAX AUC: 0.962
Villagrana-Banuelos, K.E. [36]	2022	Esophagitis vs. BE	1561	Endoscopic images	In order to classify into classes, MiniVGGNet was implemented, and after experimentation, it was tested every 50 epochs until reaching 500	VGG	No	Normal: AUC: 0.95 BE: AUC: 0.96 Esophagitis-a: AUC: 0.86 Esophagitis-b-d: AUC: 0.83

BE: Barrett’s esophagus; NDBE: non-dysplastic BE; EAC: esophageal adenocarcinoma; NBI: narrow-band imaging; CNN: convolutional neural network; GEJ: gastroesophageal junction; SCJ: squamous-columnar junction; NA: Not applicable.

**Table 2 bioengineering-10-01239-t002:** Seven studies on deep learning-assisted pathological diagnosis of BE.

Author	Year	Task	Dataset Size	Data Type	Methodology and Innovation	Model Architecture	Comparison with Experts	Result
Tomita, N. [51]	2019	Normal vs. NDBE vs. Dysplastic BE vs. EAC	123	Pathological images	A two-step attention model is proposed to extract features from high-resolution images and apply the attention mechanism for classification	CNN with attention	No	Noraml AUC: 0.751 NDBE AUC: 0.897 Dysplastic BE AUC: 0.817 EAC AUC: 0.795
Sali, R. [52]	2020	Normal vs. NDBE vs. Dysplastic BE	387	Pathological images	Compare the impact of fully supervised, weakly supervised, and unsupervised learning methods on the model	ResNet	No	MAX accuracy: 95.2%
Law, J. [53]	2021	Cell sorting	Multiple public datasets	Pathological images	An improved U-Net cell detection network is proposed, using SE(2,N) group convolution to enhance rotation invariance and optimize training	SE2-U-Net	No	Sensitivity: 92.8% F1 score 0.907
Codipilly, D.C. [54]	2021	NDBE vs. LGD vs. HGD	587	Pathological images	NA	ResNet	No	NDBE: sensitivity: 93%, specificity: 100% LGD: sensitivity: 99.2%, specificity: 95.3% HGD: sensitivity: 100%, specificity: 99.5%
Beuque, M. [55]	2021	Task 1: epithelial vs. stroma Task 2: dysplastic grade Task 3: progression of dysplasia	57	Pathological images mass spectrometry images (MSI)	MSI’s spatially resolved molecular data and H&E staining data are combined to achieve complementary lesion classification and severity grading	Grid searches + ensemble learning Convolutional Block Attention Module with Resnet50	No	Task 1: AUC 0.89 (MSI), 0.95 (H&E) Task 2: AUC 0.97 (MSI), 0.85 (H&E) Task 3: accuracy of 72% (MSI) and 48% (H&E)
Faghani, S. [56]	2022	NDBE vs. LGD vs. HGD	542	Pathological images	Whole-slide images are converted into tiles, detected using YOLO v5, then processed using a classifier model, and the results of the two models are combined	YOLO recognition and segmentation ResNet101 classification	No	NDBE F1 score: 0.91 LGD F1 score: 0.90 HGD F1 score: 1.0
Guleria, S. [57]	2021	Normal vs. NDBE vs. Dysplasia/cancer	1970 pCLE videos 897,931 biopsy patches 387 whole-slide images	PCLE endoscopic pathology images and videos	Images and videos were modeled simultaneously	NA	No	pCLE analysis: accuracy: 90% Biopsies at the patch level: accuracy: 90% Whole-slide-image-level accuracy: 94%

BE: Barrett’s esophagus; NDBE: non-dysplastic BE; EAC: esophageal adenocarcinoma; CNN: convolutional neural network; LGD: low-grade atypical hyperplasia; HGD: high-grade atypical hyperplasia; pCLE: probe-based confocal laser endomicroscopy; NA: Not applicable.

**Table 3 bioengineering-10-01239-t003:** Five studies of deep learning assisting other diagnoses of BE.

Author	Year	Task	Dataset Size	Data Type	Methodology and Innovation	Model Architecture	Comparison with Experts	Result
Fonollà, R. [63]	2019	NDBE vs. HGD	7191	Volumetric laser endomicroscopy images	Using FusionNet for VLE segmentation, features were extracted by layer histograms and gland statistics, and the model was fine-tuned with adaptive learning, data augmentation, and balanced classes	VGG-16	No	AUC: 0.96
van der Putten, J. [64]	2020	NDBE vs. HGD	140	Volumetric laser endomicroscopy images	Principal dimension encoding for VLE data is proposed, which effectively utilizes a priori information about the importance of dimensions in the image to create a lower-dimensional feature space	FusionNet/DenseNet	No	AUC: 0.93
Yang, Z. [65]	2021	Segmentation of tissue epithelium	30	OCT images	Proposed a bilateral connectivity-based neural network for in vivo human esophageal OCT layer segmentation	CE-Net (Bicon-CE)	No	Evaluate through the dice coefficient
Gehrung, M. [66]	2021	Normal vs. BE	4662	Picture of a pathological section of exfoliated cells	Proposed a classification-driven approach to analyze samples tested by Cytosponge-TFF3	VGG-16	Yes	AUC: 0.88
Waterhouse, D.J. [67]	2021	NDBE vs. EAC	715	Spectral signal	Endoscopic spectral imaging extracts vascular properties in Barrett’s esophagus to achieve high contrast	Traditional CNN	No	Sensitivity: 83.7% Specificity: 85.5%

BE: Barrett’s esophagus; NDBE: non-dysplastic BE; EAC: esophageal adenocarcinoma; HGD: high-grade atypical hyperplasia; OCT: optical coherence tomography.

**Table 4 bioengineering-10-01239-t004:** Human-machine performance comparison.

Author	Task	Data Type	Model Performance	Expert Performance
de Groof, J. [17]	BE boundary detection	Endoscopic images	Delineation scores: 35%	Delineation scores: 69%
van der Putten, J. [21]	NDBE vs. Neoplastic BE	Endoscopic images	Accuracy: 98% Sensitivity: 100% Specificity: 95%	NA
van der Putten, J. [27]	NDBE vs. Dysplastic BE	Endoscopic images	Accuracy: 87.5% Sensitivity: 92.5% Specificity: 82.5%	Accuracy: 73.0% Sensitivity: 71.8% Specificity: 74.3%
Gehrung, M. [66]	Normal vs. BE	Picture of a pathological section of exfoliated cells	Sensitivity: 72.62% Specificity: 93.13% AUC: 0.88	Sensitivity: 81.7% Specificity: 92.7%

BE: Barrett’s esophagus; NDBE: non-dysplastic BE; NA: Not applicable.

## Data Availability

Data sharing not applicable.

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
