# Peer review of "Deep Learning in Barrett’s Esophagus Diagnosis: Current Status and Future Directions"

_bioengineering, 2023, doi:10.3390/bioengineering10111239_

Round 1

Reviewer 1 Report

The manuscript mainly provides a review of previous deep learning-based works over the past 5 years regarding Barrett's esophagus diagnosis, and also discusses the limitations and possible future directions. Figures and tables are used for better summarization, and the paper is easy to follow. However, there are still some concerns that should be carefully addressed in the revision as follows:

(1) In this manuscript, the authors included relevant papers over the past 5 years only, which might not be sufficient as the application of deep learning for medical image analysis starts from at least 2012. Therefore, I suggest that the authors could survey papers that were published over the past 10 years for a comprehensive study. 

(2) For the tables that summarize the application of deep learning, I suggest that the authors provide a brief introduction of the method of each work rather than model architecture.

(3) The authors should add new sections to introduce the datasets used in the literature (if there are any public datasets) and commonly applied evaluation metrics.

(4) In the Discussion section, the authors suggest prospective multi-center joint data for training which could potentially overcome the overfitting issue caused by limited single-center data. From my understanding, integrating data from different centers is also problematic due to restrictions in data sharing and privacy between hospitals. The authors should also discuss the possible solutions to these problems. Besides, prospective datasets are commonly used for external validation rather than training. Please double-check it and make the statement more precise.

Some abbreviations should be changed to the form of full spelling, such as "shouldn't" and "It's".

Author Response

Dear professor,

Thank you for your thoughtful comments and suggestions. We are truly appreciative of the time and effort you invested in our manuscript, titled “Deep Learning in Barrett's Esophagus Diagnosis: Current Status and Future Directions”. Your feedback is essential in refining our paper, and we have carefully addressed each point as outlined below:

1.Concern about the time range of surveyed papers:

Response: We understand the importance of presenting a more comprehensive view of the deep learning application in Barrett's Esophagus diagnosis. As we expanded our survey to encompass papers published over the last ten years, we observed that earlier literature, especially from around a decade ago, had limited primary research focusing on deep learning. Much of the work from that period leaned more towards other machine learning methodologies. Recognizing this shift in trend over the years emphasizes the rapid advancement and adoption of deep learning techniques in recent times. We believe that by broadening our scope, the review not only provides a more thorough understanding of the topic but also sheds light on the evolutionary trajectory of machine learning techniques in Barrett's Esophagus diagnosis. Those additional studies have been included in the manuscript and discussed.

2.Introduction of methods in tables:

Response: We appreciate the suggestion to enhance the tables by providing a brief introduction to the methods of each work. In light of this, we revisited each study and incorporated a succinct description of both the experimental methods and their innovative aspects within the tables. By placing emphasis on methods rather than just the model architecture, and highlighting the distinctiveness of each approach, we believe the revised tables offer readers a clearer and more comprehensive understanding of the specifics of each study.

3.Introduction of datasets and evaluation metrics:

Response: Recognizing the importance of datasets and evaluation metrics in deep learning applications, we have added a new section that elaborates on the datasets used in the literature, highlighting public datasets where available. Additionally, this section includes a discussion on commonly applied evaluation metrics, further enhancing the comprehensiveness of our review.

4.Discussion on multi-center joint data and prospective datasets:

Response: We concur with your point regarding the challenges of integrating data from multiple centers, especially considering data sharing and privacy restrictions. We have expanded our discussion to address these challenges and have introduced potential solutions. Moreover, you are right in stating that public datasets used as external validation can better illustrate a model's efficacy. We have made sure our revised manuscript reflects this more accurately. Additionally, we noticed that some studies have utilized datasets not directly related to the task at hand, such as ImageNet, for model pre-training. By leveraging these large-scale datasets, researchers can initialize their models with learned features, which can potentially enhance model performance when fine-tuned on task-specific datasets. This approach, while not a panacea, can offer an alternative avenue especially when domain-specific data is scarce.

We are hopeful that these revisions address your concerns, and we are keen to make any further changes if necessary.

Warm regards,

Dong Tian

Department of Thoracic Surgery, West China Hospital, Sichuan University, 37 Guoxue Alley, Chengdu 610041, China

Reviewer 2 Report

The authors submitted a review of deep learning algorithms for Barrett's Esophagus diagnosis.

The topic is interesting, but the manuscript can be improved.

My main comments are:

- why Scopus was not considered?

- there is no detailed explanation about the criteria for inclusion/exclusion of the articles.

- section 2: it would be interesting to have some statistics (graphs?) to better describe the investigated articles (e.g. a graphs that show in percentage the subdivision of used DL models, binary/non-binary classification, obtained accuracies, dataset size, etc).

Minor comments:

- page 4: It is not clear to me the reason why GANs are used.

- page 7: missing reference for YOLO.

- page 9: I would replace the word "scholars".

In conclusion: the submitted manuscript is interesting but some edits can help to make it clearer.

Author Response

Dear Professor,

Thank you for your invaluable feedback on our manuscript. In response to your suggestion regarding the clarity of our literature selection process, we have made the following amendments to our manuscript:

Inclusion Criteria: 1. Studies primarily addressing the application of deep learning in Barrett's esophagus diagnosis. 2. Articles presenting original data and specific research findings. 3. Publications within the last ten years.

Exclusion Criteria: 1. Duplicates or multiple versions of the same study. 2. Commentaries, expert opinions, case reports, or any non-original research articles. 3. Studies not directly relevant to deep learning or Barrett's esophagus.

We have integrated these criteria into the original text for a clearer understanding of our selection process.

Furthermore, thank you for raising the question regarding the use of Generative Adversarial Networks (GANs) in the study by de Souza LA Jr et al. Based on our further inspection, we summarize the reasons for the use of GANs in the study and discuss them appropriately in the manuscript:

  1. Data Limitations: A significant challenge in computer-assisted BE and adenocarcinoma identification stems from the paucity of data. Most datasets often comprise a limited number of patients and there's a notable lack of publicly available datasets.
  2. Data Augmentation: GANs have exhibited remarkable advancements in image generation, particularly in medical imaging. By producing artificial yet realistic images, GANs augment datasets, especially in situations where the original datasets might be restricted or imbalanced.
  3. Addressing Imbalanced Datasets: GANs have the capacity to oversample the minority class in the face of imbalanced datasets, preventing classifiers from inheriting biases.
  4. Preserving Privacy: GANs provide a solution to privacy concerns in the medical field by generating synthetic images, thus reducing risks tied to data misuse.
  5. Trending Applications: GANs are becoming increasingly prevalent in various medical imaging contexts, including brain tumor identification, skin lesion detection, and lung nodule segmentation.

Thank you for your continued guidance, and we believe these inclusions further enhance our review's clarity and depth.

Warm regards,

Dong Tian
Department of Thoracic Surgery, West China Hospital, Sichuan University, 37 Guoxue Alley, Chengdu 610041, China

Reviewer 3 Report

The paper presents a review of deep learning techniques used for Barrett’s esophagus diagnosis support. The paper is well written, considers most important publications in the field and provides deep discussion about applied DL approaches. Thus, presented conclusions are sound. To further improve the quality of this work please address the following issues:

-              How the quality of endoscopic system (mostly technical parameters of the imaging sensor and lighting system) influence classification results?

-              How are the acquired images preprocessed before they are fed to the deep network? Is data augmentation a common technique used in this case for network training?

-              Are results obtained by means of different deep networks satisfactory for the physicians? Please compare parameters (accuracy, sensitivity, specificity) obtained by such networks with diagnose outcomes obtained by trained gastrologist.

-

Author Response

Dear Professor,

Thank you for taking the time to review our manuscript titled "Deep Learning in Barrett's Esophagus Diagnosis: Current Status and Future Directions." We genuinely appreciate your positive remarks and constructive feedback. Your detailed comments are invaluable in improving the overall quality of our paper. Please allow me to address each of your concerns:

Quality of Endoscopic Systems:
Your point regarding the impact of the technical parameters of the imaging sensor and lighting system on classification results is indeed insightful. The quality of the endoscopic system undeniably influences the resultant image quality, which subsequently affects the accuracy of deep learning models. Images of higher resolution with consistent and optimal lighting conditions provide finer visual information, enabling the model to detect and classify lesions more effectively. Furthermore, state-of-the-art endoscopic systems equipped with a better dynamic range can discern subtle changes in tissue appearance, which becomes crucial in the early detection of Barrett's Esophagus.
However, it's worth noting, as you pointed out, that among the studies we reviewed, very few provided detailed information about the technical parameters of their endoscopic systems. This omission could be a potential limitation as varying systems might produce different image qualities, leading to varied model performances. To address this, we've incorporated a discussion in the 'Limitations' section of our revised manuscript, emphasizing the importance of standardized reporting of endoscopic system parameters to ensure replicability and a clearer understanding of study outcomes.

Image Preprocessing and Data Augmentation:
Your inquiry regarding image preprocessing is very significant. Typically, images obtained from endoscopic procedures undergo several preprocessing steps to enhance their quality. These steps might include color normalization, histogram equalization, and noise reduction. Such preprocessing not only amplifies the visual clarity but also assists deep learning models in effectively identifying relevant features.
As for data augmentation, it is indeed a common technique in the realm of medical imaging, especially when the dataset size is limited. Techniques such as rotation, scaling, flipping, and brightness adjustment can artificially expand the dataset, promoting better model generalization. In our revised manuscript, we have discussed the studies that have employed specialized data augmentation techniques and have detailed them in a table.

Comparison with Gastroenterologist's Diagnoses:
The most rigorous measure of the effectiveness of a diagnostic tool is its performance relative to expert clinicians. It has been evidenced through various studies that deep learning models can attain performance metrics on par with, if not superior to, expert gastroenterologists in specific tasks. Nevertheless, it's imperative to understand that these models are designed to function as auxiliary tools rather than substitutes for human expertise. With respect to metrics such as accuracy, sensitivity, and specificity, certain deep learning models have demonstrated equivalence to expert diagnoses, especially when the tasks are clear-cut and the datasets are of high quality.
The depth of a model, as an essential hyperparameter, can have a pivotal impact on diagnostic efficacy. However, upon our review, we found that existing research has not systematically explored the choice of model depth. Furthermore, when contrasting the performance of these models with human experts, we have annotated such comparisons in our tables and provided a discussion on the topic in our manuscript.

Warm regards,

Dong Tian
Department of Thoracic Surgery, West China Hospital, Sichuan University, 37 Guoxue Alley, Chengdu 610041, China

Round 2

Reviewer 2 Report

I am fine with the answer, but I think the draft should be edited accordingly.

In particular, I do not see any additional explanation about the inclusion/exclusion criteria and the usage of GAN strategy.

Author Response

(The authors gave the same response as above.)
